# Photoacoustic Imaging of COVID-19 Vaccine Site Inflammation of Autoimmune Disease Patients

**DOI:** 10.3390/s23052789

**Published:** 2023-03-03

**Authors:** Janggun Jo, David Mills, Aaron Dentinger, David Chamberland, Nada M. Abdulaziz, Xueding Wang, Elena Schiopu, Girish Gandikota

**Affiliations:** 1Department of Biomedical Engineering, University of Michigan, Ann Arbor, MI 48109, USA; 2General Electric Research, Niskayuna, NY 12309, USA; 3Department of Internal Medicine, Division of Rheumatology, University of Michigan, Ann Arbor, MI 48109, USA; 4Department of Radiology, University of Michigan Medical School, Ann Arbor, MI 48109, USA; 5Division of Rheumatology, Medical College of Georgia at Augusta University, Augusta, GA 30912, USA

**Keywords:** vaccine site inflammation, COVID-19 vaccine, autoimmune disease, immunosuppressive medicine, photoacoustic imaging, Doppler ultrasound imaging

## Abstract

Based on the observations made in rheumatology clinics, autoimmune disease (AD) patients on immunosuppressive (IS) medications have variable vaccine site inflammation responses, whose study may help predict the long-term efficacy of the vaccine in this at-risk population. However, the quantitative assessment of the inflammation of the vaccine site is technically challenging. In this study analyzing AD patients on IS medications and normal control subjects, we imaged the inflammation of the vaccine site 24 h after mRNA COVID-19 vaccinations were administered using both the emerging photoacoustic imaging (PAI) method and the established Doppler ultrasound (US) method. A total of 15 subjects were involved, including 6 AD patients on IS and 9 normal control subjects, and the results from the two groups were compared. Compared to the results obtained from the control subjects, the AD patients on IS medications showed statistically significant reductions in vaccine site inflammation, indicating that immunosuppressed AD patients also experience local inflammation after mRNA vaccination but not in as clinically apparent of a manner when compared to non-immunosuppressed non-AD individuals. Both PAI and Doppler US were able to detect mRNA COVID-19 vaccine-induced local inflammation. PAI, based on the optical absorption contrast, shows better sensitivity in assessing and quantifying the spatially distributed inflammation in soft tissues at the vaccine site.

## 1. Introduction

Since the emergence of the COVID-19 pandemic, which was caused by the novel SARS-CoV-2, one of the paths to controlling the spread and minimizing the transmission, hospitalizations, and deaths related to this disease has been the successful development of vaccines in record time. The two quickly approved and widely applied mRNA vaccines, Moderna mRNA-1273 and Pfizer-BioNTech BNT162b2, are based on novel platforms. While the efficacy of these mRNA vaccines in the general population has been well reviewed, their efficacy and side effects in immune-compromised autoimmune disease (AD) patients is still being researched. This is partially because AD patients on immunosuppressive (IS) medications have been excluded from the pivotal vaccine trials before approval. The CDC estimates that at least 9 million Americans belong to this at-risk category [1]. Recent data confirm that AD patients on IS medications have a much higher prevalence of COVID-19 and are at increased risk of developing severe conditions [2,3]. Several recent studies have also demonstrated that AD patients on IS medications experience reduced efficacy with respect to the mRNA COVID-19 vaccines [4,5,6,7]. A study by researchers at Johns Hopkins University found that fully vaccinated immunocompromised people accounted for about 44% of the breakthrough cases that led to hospitalization, and vaccinated immunocompromised people were 485 times more likely than other vaccinated people to be hospitalized or die from this disease [1].

Both the Moderna and Pfizer-BioNTech mRNA vaccines are administered via an intramuscular route and function by triggering vaccine site inflammation. The mRNA-based vaccines actively induce both humoral (B-cell) and cell-mediated (T-cell) immune responses. At the vaccine injection site in the muscle, the myocytes take up the lipid nanoparticles and then release the mRNAs into the cytoplasm for translation into spike-proteins. The immune system sets off complex innate immune reactions that are crucial for triggering the strong, antigen-specific acquired immune responses that are necessary to protect against the disease [8]. Hence, it is reasonable to hypothesize that the local inflammation triggered by vaccines can be a predictive sign of desirable vaccine responses in the long term [8,9]. The strong correlation between the local inflammation and magnitude of vaccine immunogenicity has been studied for more traditional vaccines such as that for hepatitis B, and such studies have confirmed that the potency of an adjuvant system in terms of inducing an innate response is positively correlated with the magnitude of the adaptive response [10]. Although this correlation has not been established for the novel COVID-19 mRNA vaccines, our observation in rheumatology clinics indicates that AD patients on IS medications show variable vaccine site inflammation responses, which could be an early reflection of AD patients’ poor responses to COVID-19 mRNA vaccines.

Assessing and quantifying the inflammation of the vaccine site, however, is technically challenging. Currently, there is no clinically accepted protocol for the quantitative assessment of the injection site’s inflammation after vaccination. Temperature, pain, redness, and swelling are non-specific and subjective, depending on factors such as the presence of other medical conditions and patients’ perception of pain. Conventional imaging techniques such as X-ray imagery, computed tomography (CT), and magnetic resonance imaging (MRI) may detect such pathologies but mainly rely on anatomical changes that are not sensitive to mild and early inflammation [11]. Dynamic contrast-enhanced (DEC) MRI, by examining the uptake of gadolinium over time, provides better sensitivity for monitoring mild inflammation [12,13]. However, the high cost, limited access, and involvement of contrast agents render DEC-MRI difficult to use as a point-of-care tool. Ultrasound imaging (US) is widely available, quick, inexpensive, and not associated with radiation exposure, offering a point-of-care tool for many clinical settings such as rheumatology clinics [14,15]. Besides presenting anatomical tissue changes, US is also sensitive in terms of identifying hypervascularity by detecting blood flow based on the Doppler effect. US is particularly suitable for imaging soft tissues within a few centimeters, wherein high-frequency probes can be used to present spatial details on the order of 100 µm. The sensitivity of Doppler US in detecting vaccine site inflammation, however, is limited. This is due to the fact that Doppler US is more sensitive to the rapid blood flow in relatively large vessels. In contrast, the slow blood flow in smaller capillaries, which are more pathologically relevant to mild inflammation in soft tissues, may be neglected [16].

In this study, by leveraging the high sensitivity of the emerging photoacoustic imaging (PAI) technology for detecting soft tissue inflammation, we quantitatively studied the local vaccine site inflammation 24 h after COVID-19 vaccinations for AD patients on IS medications. Our recently developed protocol based on PAI and US dual-modality imaging renders a promising new tool for detecting and quantifying soft tissue inflammation [17,18,19,20,21,22,23,24,25,26,27,28]. By working at the optical wavelengths leading to strong optical absorption by hemoglobin, PAI can identify and characterize inflammation based on the detection of hyperemia (i.e., enhanced blood volume), which is a physiological hallmark reflecting the increased metabolic activity of inflammatory tissues. Unlike Doppler US, which detects blood vessels based on blood flow measurements, PAI presents the contrast of optical absorption proportional to blood volume. Hence, PAI possesses the ability to assess vascularity in all dimensions and with any flow velocities. Therefore, PAI has higher sensitivity for detecting increased levels of microvascular and capillary blood, which are two forms of blood that are less accounted for by Doppler US yet more relevant to inflammation [27,28]. As presented in this work, our initial study on subjects receiving COVID-19 vaccines demonstrated that PAI can sensitively map the spatially distributed microvascular activities related to vaccine site inflammation. Compared to the imaging results from the normal control subjects, the AD patients on IS therapies showed a statistically significant difference in vaccine site inflammation.

## 2. Materials and Methods

In this pilot study, we scanned a total of 15 subjects, including 6 AD patients on IS medications and 9 normal control subjects who were not on any IS medications. The AD patients were recruited from the Rheumatology Clinic at the University of Michigan. A board-certified rheumatologist confirmed both the pathological conditions of the AD patients and the non-AD medical status of the control subjects. All procedures for human subjects used in this study were approved by the Institutional Review Board (IRB) of the University of Michigan Medical School (HUM00003693).

For each subject, the inflammation of the vaccine site in the deltoid was imaged at around 24 h following mRNA COVID-19 vaccination. PAI was performed using a photoacoustic–ultrasound dual system based on LED light source (AcousticX, Cyberdyne, Tsukuba, Japan). A photograph of this imaging system and the schematic regarding the scanning of vaccine site inflammation are shown in Figure 1. The imaging probe has two LED array bars located on both sides of an ultrasound array transducer, forming an angle of 45° with respect to the center plane of the probe. The pulsed LED source provides light energy of 400 µJ per pulse at 850 nm, with a repetition rate of 4 kHz and a pulse duration of 35 ns. After processing the photoacoustic signals using frequency band pass filter, log compression, and time gain compensation, the images were reconstructed using the delay-and-sum method and then displayed in pseudo-color. The photoacoustic images obtained from over 384 light pulses were averaged to enhance the signal-to-noise ratio, leading to an imaging frame rate of approximately 10 Hz. Besides acquiring and displaying the photoacoustic images in pseudo-color, this imaging system acquires B-mode ultrasound images from the same imaging plane using the same probe, which also occurs in real time. When working with a 7 MHz linear array transducer with 128 elements (Cyberdyne, Tsukuba, Japan), this LED-based PAI system offers a spatial resolution of 310 μm laterally and 250 μm axially, as well as an image depth up to 30 mm. The details of this system and its applications for the imaging of soft tissue inflammation have been reported in previous studies [26,27,28,29,30].

To compare these results to the PAI results, the same vaccination site of each subject was also scanned using an ultrasound unit (Vivid^TM^ E95, GE HealthCare, Chicago, IL, USA) modified such that PAI could be incorporated, operating with an L8-18i-D probe. To be used as a reference, the contralateral side (mirror opposite) in the opposite arm of each subject was also imaged by the same imaging systems with the same settings.

## 3. Results

We were able to include a total of 15 subjects: 6 with a known AD and 9 without an AD. The demographics (age at the time of the study, gender, ethnicity, and BMI) and clinical characteristics (AD diagnosis, disease duration, and IS medications) are illustrated in Table 1. For patients with myositis, we determined whether the disease was active (Y), which was ascertained by through the observation of increased muscle inflammation and weakness.

An example imaging result from an AD patient is shown in Figure 2. Compared to the reference site (for which there was no vaccine injection), the PAI image at the vaccine site shows slightly enhanced signals, primarily in subcutaneous fat and at the interface between fat and muscle. This mild increase in vascularity due to inflammation at the vaccine site detected by PAI was confirmed by the Doppler US image. Compared to the one from the reference site, the Doppler US image at the vaccine site also shows very mild activity at the fat–muscle interface. An example imaging result from a normal control subject is shown in Figure 3. Compared to the reference site (for which there was no vaccine injection), the PAI image at the vaccine site shows strongly enhanced signals indicating spatially distributed hyperemia in the fat and muscle tissues. The Doppler US confirmed the active inflammation at the vaccine site detected by PAI. Compared to the one at the reference site and the one from the AD patient in Figure 2c, the Doppler US image in Figure 3c shows strong activities in both fat and muscle via a few scattered large vessels.

To quantitatively evaluate the local inflammation induced by the mRNA vaccine and compare the results from the two subject groups, we calculated both the density and the average intensity of the colored pixels around the vaccine site in the pseudo-color PAI images. When the density of the colored pixels was quantified, the total area of the colored pixels was divided by the total area of the target tissue in the PA image. When the intensity of the colored pixels was quantified, the average intensity of all the colored pixels was calculated. Both of these two measurements can be utilized to quantify the severity of hyperemia, as discussed in our previous publication [24]. For each subject, these two measurements were also normalized by those at the reference site, which helped reduce the influence from the variables such as the skin color and BMI of the subject. The normalized measurements of vaccine site inflammation of the AD patients on IS (*n* = 6) are compared to those from normal control subjects who were not on IS (*n* = 9) are shown in Figure 4. Both the density and the average intensity of the colored pixels at the vaccine site are stronger for the control subjects when compared to the AD patients, indicating significantly reduced inflammatory responses to the mRNA vaccine in the AD patients under IS treatments at the time of vaccination. Statistical analyses were performed using two-sample Wilcoxon tests to compare the quantitative measurements from the two subject groups. The calculated p values were *p* = 0.0056 and *p* = 0.0113 for the comparison studies based on the density and the average intensity of the colored pixels in the PA images, respectively.

## 4. Discussion and Conclusions

In this study on AD patients on IS medications and normal control subjects, we quantitatively evaluated the local inflammation of the vaccine site at 24 h after mRNA COVID-19 vaccines were administered. The inflammation of the vaccine site was imaged by both emerging PAI and clinical Doppler US. The imaging results from the vaccination site of each subject were also compared to those from the contralateral site in the subject’s opposite arm. Both PAI and Doppler US can sensitively map the spatially distributed microvascular activity presenting vaccine site inflammation. Compared to Doppler US, which could detect the inflammation of the vaccine site but only showed a few vessels, PAI presented the spatially distributed microvascular activities at the vaccine site. We hypothesize that information on volumetric hyperemia obtained through PAI may provide a sensitive and quantitative assessment of the overall severity of inflammation. Compared to the imaging results of normal control subjects, the AD patients on IS therapy showed a statistically significant reduction in the photoacoustic signal at the vaccine site, suggesting that patients with AD also experience local inflammation after being administered an mRNA vaccine but not in as clinically evident of a manner compared to non-immunosuppressed individuals.

As a next step in this study, we will explore whether there is a direct correlation between the features extracted from photoacoustic images at the vaccine site and the degree of protection provided by mRNA vaccination. Once this correlation can be established, we may be able to develop a cost-efficient and point-of-care-capable imaging procedure that can enable the early prediction of the efficacy of mRNA COVID-19 vaccines in AD patients on IS medications, thereby facilitating early intervention and dose adjustment to achieve the effective and prolonged protection of this at-risk population.

## Figures and Tables

**Figure 1 sensors-23-02789-f001:**
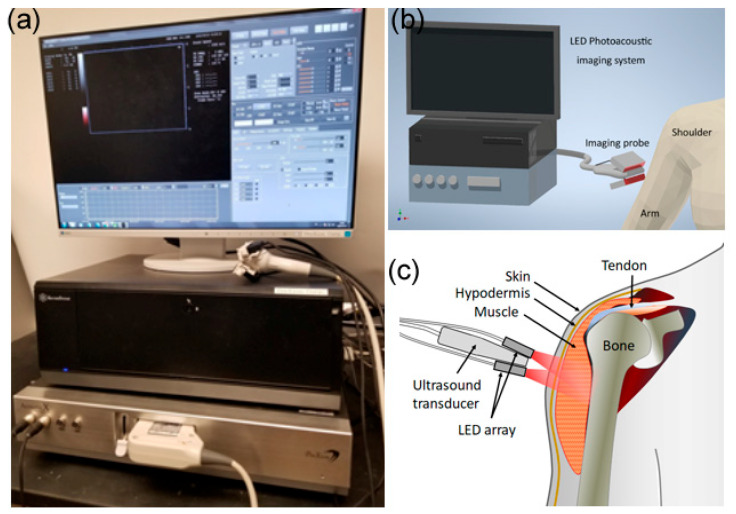
Photoacoustic imaging of vaccine site inflammation: (**a**) Photograph of the LED-based photoacoustic imaging system; (**b**,**c**) Schematic for scanning soft-tissue inflammation at the vaccination site.

**Figure 2 sensors-23-02789-f002:**
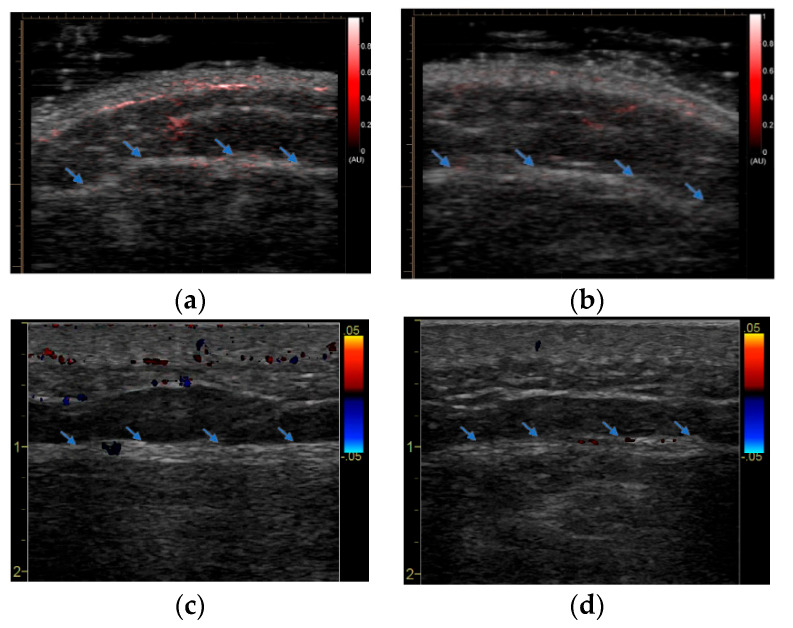
PAI and Doppler US imaging of an AD patient (on IS) obtained around 24 h after injection of mRNA vaccine: (**a**) PAI image of the injection site; (**b**) PAI image of the reference site (the contralateral site in the other arm). In (**a**,**b**), the pseudo–color PAI images are superimposed on the gray–scale US images. (**c**) Doppler US image of the same injection site acquired at the same time as PAI was conducted. (**d**) Doppler US image of the same reference site acquired at the same time as PAI was conducted. The blue arrows mark the interface between fat and muscle.

**Figure 3 sensors-23-02789-f003:**
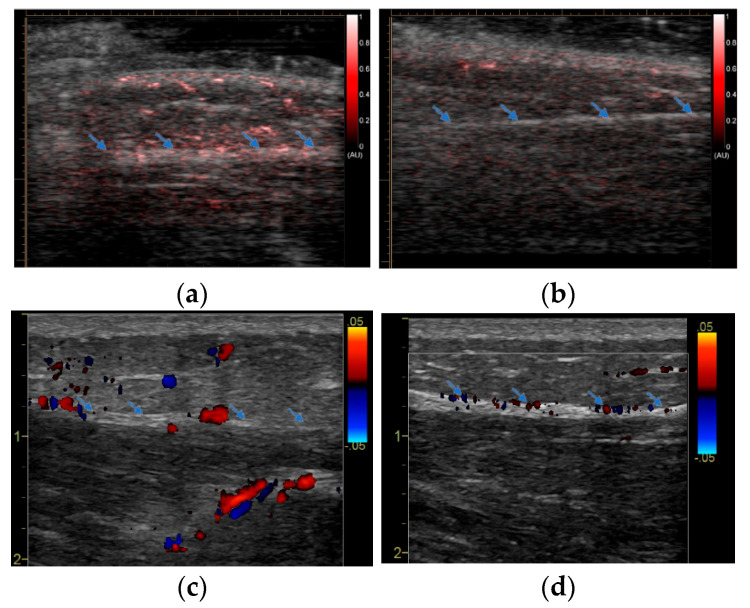
PAI and Doppler US imaging of a control subject (not on IS) conducted around 24 h after mRNA vaccine administration: (**a**) PAI image of the injection site; (**b**) PAI image of the reference site (the contralateral site in the other arm); (**c**) Doppler US image of the same injection site acquired at the same time PAI was conducted; (**d**) Doppler US image of the same reference site acquired at the same time PAI was conducted. The blue arrows mark the interface between fat and muscle.

**Figure 4 sensors-23-02789-f004:**
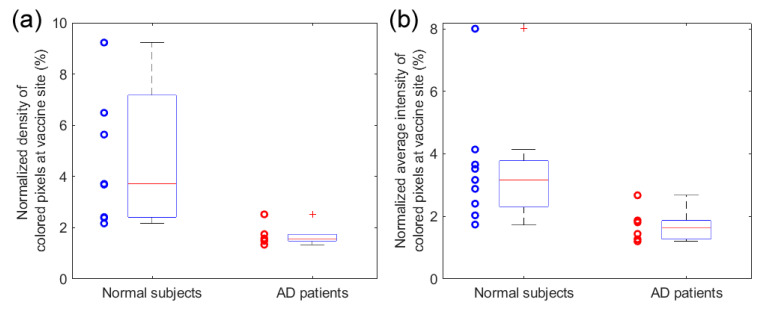
Statistical analyses of the PAI results quantifying the inflammation at the vaccination site. The results from the group of 9 normal control subjects (not on IS) are compared to those of 6 AD patients (on IS): (**a**) the normalized density of the colored pixels at the vaccine site in pseudo-color PA images from the two groups; (**b**) the normalized average intensity of the colored pixels at the vaccine site in the pseudo-color PA images from the two groups. Two-sample Wilcoxon tests were conducted to compare the quantitative measurements from the two subject groups, leading to values of *p* = 0.0056 and *p* = 0.0113 for the comparison studies of (**a**,**b**), respectively.

**Table 1 sensors-23-02789-t001:** Demographics and clinical characteristics of the study subjects. BMI = body mass index; JDM = juvenile dermatomyositis; DM = dermatomyositis; RA = rheumatoid arthritis; IMNM = immune mediated necrotizing myositis; GC = glucocorticoids; HCQ = hydroxychloroquine; MTX = methotrexate; MMF = mycophenolate mofetil; RTX = rituximab.

Study#	Age	Ethnicity/Gender	BMI	AD Diagnosis	Disease Duration (Years)	MyositisActive	Treated
1	34	White/Female	21				
2	52	White/Male	27.6				
3	45	White/Female	21				
4	49	Indian/Male	23.5				
5	36	Black/Female	36.1	JDM	15	N	GC, HCQ
6	51	Indian/Male	24.4				
7	63	White/Male	26.4	DM	9	N	MTX
8	64	White/Female	41.5	DM	33	N	GC, MMF, Ustekinumab
9	66	White/Female	22.2	DM	1	Y	GC, HCQ, RTX
10	57	White/Female	23	RA	19	Y	MTX
11	68	White/Female	31.8	IMNM	4	Y	MMF
12	24	White/Female	31.8				
13	85	White/Female	25.2				
14	86	White/Male	30.7				
15	79	White/Male	27.2				

## Data Availability

The data that support the finding of this study are available from the corresponding author on request.

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
