# Peer review of "Photoacoustic Imaging of COVID-19 Vaccine Site Inflammation of Autoimmune Disease Patients"

_sensors, 2023, doi:10.3390/s23052789_

Round 1

Reviewer 1 Report

The manuscript entitled "Photoacoustic Imaging of COVID-19 Vaccine Site Inflammation of Autoimmune Disease Patients" by Janggun Jo et al. provides interesting insights into the feasibility of PAI as a sustainable tool to predict the degree of protection of patients undergoing immunosuppressive treatment. The overall work is interesting, well designed and clearly up to date as it combines the recent outbreak and a thriving tool like PAI. And their combination is clever, as it uses the main hallmark of PAI to provide information on vascularity, while also meeting the need of COVID19 for accessible solutions. The statistical relevance of the experimental work is limited, but the results are so promising and well-founded that they certainly deserve to be brought to the attention of the scientific and medical community. Furthermore, this manuscript is well written and it was a real pleasure to read it. However, prior to publication in MDPI Sensors, I recommend the following minor revision:

Table 1 Please add a title for the last column. I'm not sure I fully understand the meaning of the column named Active.

Caption to Figure 3: mRAN instead of mRNA

"... we calculated both the density and the average intensity of the colored pixels around the vaccine site in the pseudo-color PAI images." The notion of density of the colored pixels is rather unclear. Does it originate from a threshold applied to the intensity of the photoacoustic signals? Please elaborate more on this concept.

"The volumetric hyperemia information from PAI enabled a sensitive and quantitative assessment of the overall severity of inflammation." This manuscript does not prove this claim, which would have required a quantitative comparison between data obtained from PAI and those from an established probe of inflammation such as Doppler US. May I recommend mitigating this statement, such as "We hypothesize that information on volumetric hyperemia from PAI may provide a sensitive, quantitative assessment of the overall severity of inflammation."

"Compared to the imaging results from the normal control subjects, AD patients on IS therapies show statistically significant reduction in vaccine site inflammation, indicating that AD patients also experience local inflammation after mRNA vaccination but not as clinically apparent when compared to non-immunosuppressed individuals." As a consequence of the comment above, may I also recommend that this statement be changed to read "Compared to the imaging results of normal control subjects, AD patients on IS therapy show a statistically significant reduction in the photoacoustic signal, suggesting that patients with AD also experience local inflammation after mRNA vaccination but not as clinically evident compared to non-immunosuppressed individuals."

"As the next step of this study, we will explore whether there is a direct correlation between the extent of local inflammatory response elicited at the vaccine site and the degree of protection provided by the mRNA vaccination." This statement should also be rephrased as "As a next step in this study, we will explore whether there is a direct correlation between features extracted from photoacoustic images at the vaccine site and the degree of protection provided by mRNA vaccination."

Once these points are addressed, in my opinion this manuscript will be ideal for publication in MDPI Sensors.

Author Response

Answer to Review Comments and Suggestions for Authors

 Reviewer 1

The manuscript entitled "Photoacoustic Imaging of COVID-19 Vaccine Site Inflammation of Autoimmune Disease Patients" by Janggun Jo et al. provides interesting insights into the feasibility of PAI as a sustainable tool to predict the degree of protection of patients undergoing immunosuppressive treatment. The overall work is interesting, well designed and clearly up to date as it combines the recent outbreak and a thriving tool like PAI. And their combination is clever, as it uses the main hallmark of PAI to provide information on vascularity, while also meeting the need of COVID19 for accessible solutions. The statistical relevance of the experimental work is limited, but the results are so promising and well-founded that they certainly deserve to be brought to the attention of the scientific and medical community. Furthermore, this manuscript is well written and it was a real pleasure to read it. However, prior to publication in MDPI Sensors, I recommend the following minor revision:

Table 1 Please add a title for the last column. I'm not sure I fully understand the meaning of the column named Active.

We added the title for the last column as “Treated”.  In Table 1, the ‘Active’ describe if the patients have myositis, which is an inflammatory muscle disease, at the time of the vaccination injection. The active myositis may have possible relevant to Covid vaccination. And, the column was included in the table. We added the disease name in Table 1 and the sentence, “For patients with Myositis, we captured whether the disease was active (Y): increased muscle inflammation and weakness.” in the Results.

Caption to Figure 3: mRAN instead of mRNA

We corrected the error.

"... we calculated both the density and the average intensity of the colored pixels around the vaccine site in the pseudo-color PAI images." The notion of density of the colored pixels is rather unclear. Does it originate from a threshold applied to the intensity of the photoacoustic signals? Please elaborate more on this concept.

The density indicates the area of PA image. We used these two parameters of density and intensity in the previous study, [Jo.J., Scientific Reports, 7, 15026]. The density of the pseudo-color pixels shows how wide the detected area is, by the value of colored pixels/ total pixels, when the intensity shows the strength of the PA images. We added the reference and the explanation, “When quantify the density of the colored pixels, the total area of colored pixels was divided by the total area of the target tissue in the PA image. When quantify the intensity of the colored pixels, the average intensity of all colored pixels was calculated. Both of these two measurements can be utilized to quantify the severity of hyperemia, as discussed in our previous publication.” in the manuscript.

"The volumetric hyperemia information from PAI enabled a sensitive and quantitative assessment of the overall severity of inflammation." This manuscript does not prove this claim, which would have required a quantitative comparison between data obtained from PAI and those from an established probe of inflammation such as Doppler US. May I recommend mitigating this statement, such as "We hypothesize that information on volumetric hyperemia from PAI may provide a sensitive, quantitative assessment of the overall severity of inflammation."

We edited the sentences in Discussion and Conclusions.

 "Compared to the imaging results from the normal control subjects, AD patients on IS therapies show statistically significant reduction in vaccine site inflammation, indicating that AD patients also experience local inflammation after mRNA vaccination but not as clinically apparent when compared to non-immunosuppressed individuals." As a consequence of the comment above, may I also recommend that this statement be changed to read "Compared to the imaging results of normal control subjects, AD patients on IS therapy show a statistically significant reduction in the photoacoustic signal, suggesting that patients with AD also experience local inflammation after mRNA vaccination but not as clinically evident compared to non-immunosuppressed individuals."

We edited the sentences in Discussion and Conclusions.

 "As the next step of this study, we will explore whether there is a direct correlation between the extent of local inflammatory response elicited at the vaccine site and the degree of protection provided by the mRNA vaccination." This statement should also be rephrased as "As a next step in this study, we will explore whether there is a direct correlation between features extracted from photoacoustic images at the vaccine site and the degree of protection provided by mRNA vaccination."

We edited the sentences in Discussion and Conclusions.

 Once these points are addressed, in my opinion this manuscript will be ideal for publication in MDPI Sensors.

We appreciate your review comments and recommendations.

Reviewer 2 Report

I have the following questions

1Why choose 850 nm laser for photoacoustic imaging?

2Why does Figs 2 (d) and 3 (d) look like mirror images compared to other images?

3Whether there is compensation for the scattering and attenuation of optical signal in the tissue during quantitative analysis

4What is the basis for calculating both the density and the average intensity of the colored pixels around the vaccine site in the pseudo-color PAI images

Author Response

Answer to Review Comments and Suggestions for Authors

Reviewer 2

I have the following questions:

1、Why choose 850 nm laser for photoacoustic imaging?

With PA imaging, we detected hyperemia caused by inflammation. At the 850-nm wavelength, deoxygenated hemoglobin has lower optical absorption compared to oxygenated hemoglobin. Therefore, the increase in PA signal intensity in muscles and joints should come from the higher total hemoglobin concentration.

2、Why does Figs 2 (d) and 3 (d) look like mirror images compared to other images?

During locating the images, the two images may be flipped. The images were corrected.

3、Whether there is compensation for the scattering and attenuation of optical signal in the tissue during quantitative analysis?

We compensated the PA imaging gains to reduce the noisy in the PA images of the patients’ deltoid muscles. They were imaged by the commercial LED-based PAI system offering 384 times averaged PA images. The deltoid muscle are relatively simple soft-tissue. In the gain control function in the system, we can pick the PA images with improved SNR.

4、What is the basis for calculating both the density and the average intensity of the colored pixels around the vaccine site in the pseudo-color PAI images?

We used these two parameters of density and intensity in the previous study, [Jo.J., Scientific Reports, 7, 15026]. The density of the pseudo-color pixels shows how wide the detected area is, by the value of colored pixels/ total pixels, when the intensity shows the strength of the PA images. We added the reference and the explanation, “When quantify the density of the colored pixels, the total area of colored pixels was divided by the total area of the target tissue in the PA image. When quantify the intensity of the colored pixels, the average intensity of all colored pixels was calculated. Both of these two measurements can be utilized to quantify the severity of hyperemia, as discussed in our previous publication.” in the manuscript.